# Posterior Lissencephaly Associated with Subcortical Band Heterotopia Due to a Variation in the *CEP85L* Gene: A Case Report and Refining of the Phenotypic Spectrum

**DOI:** 10.3390/genes12081208

**Published:** 2021-08-05

**Authors:** Gianluca Contrò, Alessia Micalizzi, Sara Giangiobbe, Stefano Giuseppe Caraffi, Roberta Zuntini, Simonetta Rosato, Marzia Pollazzon, Alessandra Terracciano, Manuela Napoli, Susanna Rizzi, Grazia Gabriella Salerno, Francesca Clementina Radio, Marcello Niceta, Elena Parrini, Carlo Fusco, Giancarlo Gargano, Renzo Guerrini, Marco Tartaglia, Antonio Novelli, Orsetta Zuffardi, Livia Garavelli

**Affiliations:** 1Medical Genetics Unit, Azienda USL-IRCCS di Reggio Emilia, 42123 Reggio Emilia, Italy; gianluca.contro@ausl.re.it (G.C.); StefanoGiuseppe.Caraffi@ausl.re.it (S.G.C.); roberta.zuntini@ausl.re.it (R.Z.); simonetta.rosato@ausl.re.it (S.R.); marzia.pollazzon@ausl.re.it (M.P.); 2Laboratory of Medical Genetics, Translational Cytogenomics Research Unit, Bambino Gesù Children’s Hospital, IRCCS, 00165 Rome, Italy; alessia.micalizzi@opbg.net (A.M.); alessandra.terracciano@opbg.net (A.T.); antonio.novelli@opbg.net (A.N.); 3Clinical Genomics, Medical Genetics Service, San Raffaele Hospital, 20132 Milan, Italy; Giangiobbe.Sara@hsr.it; 4Neuroradiology Unit, Azienda USL-IRCCS di Reggio Emilia, 42123 Reggio Emilia, Italy; manuela.napoli@ausl.re.it; 5Child Neurology and Psychiatry Unit, Azienda USL-IRCCS di Reggio Emilia, 42123 Reggio Emilia, Italy; susanna.rizzi@ausl.re.it (S.R.); GraziaGabriella.Salerno@ausl.re.it (G.G.S.); carlo.fusco@ausl.re.it (C.F.); 6Genetics and Rare Diseases Research Division, Bambino Gesù Children’s Hospital, IRCCS, 00165 Rome, Italy; fclementina.radio@opbg.net (F.C.R.); marcello.niceta@opbg.net (M.N.); marco.tartaglia@opbg.net (M.T.); 7Pediatric Neurology, Neurogenetics and Neurobiology Unit and Laboratories, Meyer Children’s Hospital, University of Florence, 50139 Florence, Italy; elena.parrini@meyer.it (E.P.); renzo.guerrini@meyer.it (R.G.); 8Neonatal Intensive Care Unit, Azienda USL-IRCCS di Reggio Emilia, 42123 Reggio Emilia, Italy; giancarlo.gargano@ausl.re.it; 9Unit of Medical Genetics, Department of Molecular Medicine, University of Pavia, 27100 Pavia, Italy; orsetta.zuffardi@unipv.it

**Keywords:** *CEP85L*, lissencephaly 10, posterior lissencephaly, double-cortex, abnormalities of cortical development, whole exome sequencing, donor splice site

## Abstract

Lissencephaly describes a group of conditions characterized by the absence of normal cerebral convolutions and abnormalities of cortical development. To date, at least 20 genes have been identified as involved in the pathogenesis of this condition. Variants in *CEP85L*, encoding a protein involved in the regulation of neuronal migration, have been recently described as causative of lissencephaly with a posterior-prevalent involvement of the cerebral cortex and an autosomal dominant pattern of inheritance. Here, we describe a 3-year-old boy with slightly delayed psychomotor development and mild dysmorphic features, including bitemporal narrowing, protruding ears with up-lifted lobes and posterior plagiocephaly. Brain MRI at birth identified type 1 lissencephaly, prevalently in the temporo–occipito–parietal regions of both hemispheres with “double-cortex” (Dobyns’ 1–2 degree) periventricular band alterations. Whole-exome sequencing revealed a previously unreported de novo pathogenic variant in the *CEP85L* gene (NM_001042475.3:c.232+1del). Only 20 patients have been reported as carriers of pathogenic *CEP85L* variants to date. They show lissencephaly with prevalent posterior involvement, variable cognitive deficits and epilepsy. The present case report indicates the clinical variability associated with *CEP85L* variants that are not invariantly associated with severe phenotypes and poor outcome, and underscores the importance of including this gene in diagnostic panels for lissencephaly.

## 1. Introduction

Lissencephaly (LIS), literally “smooth brain”, is a condition characterized by decreased gyral and sulcal development of the cerebral surface. It is caused by abnormal neuronal migration during the first weeks of gestation, resulting in agyria (absent gyration), pachygyria (wide gyri) or a mix of agyria/pachygyria. Cerebral cortex structure shows abnormally thick and disorganized layers associated with diffuse neuronal heterotopia, dysmorphic ventricles, and often hypoplasia of the corpus callosum [1,2]. Subcortical band heterotopia (SBH), which represents the less severe end of the spectrum of these malformations, consists of a band of heterotopic neurons located beneath the cortex and separated from it by a thin layer of white matter. Pachygyria and SBH can occur simultaneously [3,4]. Di Donato et al. in 2017 reviewed the neuroradiological patterns of lissencephaly and proposed a classification based on the severity spectrum (agyria, pachygyria and SBH), on its topographical distribution (e.g., anterior, posterior and diffuse) and on the presence of associated anomalies [5]. Microlissencephaly represents a separate subgroup and is defined as a combination of lissencephaly (usually in the form of agyria or pachygyria) with severe congenital microcephaly (head circumference at birth ≥ 3 standard deviations below average) [6].

Lissencephaly comprises both isolated and syndromic forms and is a genetically heterogeneous disorder, including both single gene variants and locus deletions [7]. In 2018, Di Donato et al. conducted a systematic analysis of the genetic basis of lissencephaly in a large cohort of patients. They evaluated 17 known associated genes as well as 17p13.3 deletion, responsible for the syndromic form of Miller–Dieker syndrome. Diagnostic yield was 81%, with the four most frequent genes—*LIS1 (PAFAH1B1)*, *DCX*, *TUBA1A*, and *DYNC1H1*—accounting for 69% of all cases. The authors identified a consistent phenotypic spectrum, defined primarily by brain imaging features, especially by the anterior-to-posterior gradient. The identified genes are involved in neuronal migration pathways, with many of them encoding centrosomal proteins required for microtubule cytoskeleton organization [8].

In 2020, Kodani et al. described seven individuals from seven families with variants in *CEP85L* with strikingly similar radiographical and clinical features [9]. In the same year, variants in the *CEP85L* gene were described in a small group of 13 patients with sporadic predominant posterior lissencephaly, outlining some common clinical features in addition to the cerebral malformation [10].

Here, we report the case of a 3-year-old boy with a de novo splicing variant in *CEP85L.* He presented with mild neurodevelopmental delay and a brain magnetic resonance imaging (MRI) showing temporo–parieto–occipital prevalent lissencephaly with subcortical band heterotopia, a combination not reported before in *CEP85L* patients. We compare this with other known cases in order to further define the clinical characteristics of this condition (lissencephaly type 10, MIM #618873), in particular, regarding the MRI, electroencephalographic (EEG) and neurodevelopment aspects. Moreover, it suggests the importance of including this gene among those analyzed in patients with posterior lissencephaly.

## 2. Materials and Methods

### 2.1. Whole-Exome Sequencing

Genomic DNA was extracted from peripheral blood leucocytes with QIAgen columns (QIAsymphony DNA minikit, Qiagen, Hilden, Germany) according to the manufacturer’s instructions. Concentration and purity of DNA samples were quantified by ND-1000 spectrophotometer (NanoDrop; Thermo Scientific, Waltham, MA, USA) and by FLx800 Fluorescence Reader (BioTek, Winooski, VT, USA). The proband and his parents have provided written informed consent for molecular analysis.

Trio-based whole-exome sequencing (WES) was performed on genomic DNA by using the Twist Human Core Exome Kit (Twist Bioscience, South San Francisco, CA, USA) according to the manufacturer’s protocol on a NovaSeq6000 platform (Illumina, San Diego, CA, USA). The reads were aligned to human genome build GRCh37/UCSC hg19. The Dragen Germline Enrichment application of BaseSpace (Illumina, San Diego, CA, USA) and the Geneyx Analysis (Knowledge-Driven NGS Analysis tool powered by the GeneCards Suite, Wilmington, DE, USA) were used for the variant calling and annotating variants, respectively. Sequence data were carefully analyzed and the presence of all suspected variants was checked in public databases (dbSNP [11], 1000 Genomes Project [12], EVS [13], ExAC [14], gnomAD [15]). Exome sequencing data filtering was performed to identify protein-altering, putative rare recessive homozygous, compound heterozygous, and pathogenic or likely pathogenic heterozygous variants with an allele frequency < 1%, according to ExAC’s overall frequency, that result in a change in the amino acid sequence (i.e., missense/nonsense), or that reside within a canonical splice site. The variants were evaluated by VarSome [16] and categorized in accordance with the ACMG recommendations [17]. Variants were examined for coverage and Qscore (minimum threshold of 30) and visualized by the Integrative Genome Viewer (IGV). Simultaneously WES raw data were processed and analyzed using an in-house implemented pipeline previously described [18,19,20] mainly based on the GATK Best Practices. Briefly, the UCSC GRCh37/hg19 genome assembly was used for reads alignment by means of the BWA-MEM tool [21] and variant calling used HaplotypeCaller (GATK v3.7) [22]. Variants/genes functional annotation was made by SnpEff v.4.3 and dbNSFP v.3.5, and various clinical databases (OMIM, ClinVar, HGMD). In silico prediction of the impact of variants was performed by Combined Annotation Dependent Depletion (CADD) v.1.4, Mendelian Clinically Applicable Pathogenicity (M-CAP) v.1.0, and Intervar v.2.0.1. By filtering against an in-house population-matched database (~2500 exomes) and public databases (dbSNP150 and gnomAD V.2.0.1), the analysis was focused on high-quality rare variants affecting coding sequences and adjacent intronic regions. For variant prioritization, we took advantage of an in-house developed scoring system integrating various variant functional annotation (by means of SnpEff), variant clinical impact according to ACMG criteria (by means of InterVar), in silico prediction of variant functional effect on protein function (by means of CADD, M-CAP, S-CAP, Spidex), population frequency for the variant in different databases (gnomAD and internal one), gene tolerance to mutations (GDI and RVIS tools from dbNSP3), and genotype–phenotype correlation (by means of Phenolyzer). Candidate splicing variants were further characterized in silico using the Human Splicing Finder [23], NNSPLICE [24], NetGene2 [25], MaxEntScan [26], and SPiCEv2.1.5 [27].

### 2.2. mRNA Analysis

Total RNA was isolated from peripheral blood lymphocytes of the patient and his parents using RNeasy Mini Kit (Qiagen, GmbH, Hilden, Germany). A total of 500 ng RNA was reverse-transcribed into cDNA using random hexamer primers and a Transcriptor First Strand cDNA Synthesis kit (Roche Diagnostics, Indianapolis, IN, USA). PCR for evaluation of the variant effect on mRNA was performed using forward primer and reverse primer mapped on exon 1 and 3, respectively; β-actin was used as internal control (Appendix A). PCR products were analyzed by standard gel electrophoresis and purified by gel extraction. Finally, Sanger sequencing was carried out using Big Dye Terminator v1.1 Cycle Sequencing Kit (Applied Biosystems, Warrington, Cheshire, UK) and run on an ABI 3500 Dx Genetic Analyzer (Applied Biosystems).

## 3. Results

### 3.1. Clinical Case Presentation

Our patient is the first son of healthy non-consanguineous parents, first evaluated at birth at the neonatology department due to prenatal evidence of a borderline ventriculomegaly at 40 gestational weeks. He was born from spontaneous delivery, with a birth weight of 3285 g, length of 50 cm, head circumference of 33 cm and an Apgar score of 9 at 1 min and 10 at 5 min. At the last check-up at 2 years and 9 months, the child was 92.5 cm tall (50th percentile), weighed 15 kg (75th percentile) and had a head circumference of 49 cm (50th–75th percentile). Physical examination revealed bitemporal narrowing, slightly protruding ears with up-lifted lobes, posterior plagiocephaly, large first toe and no other major abnormalities (Figure 1A–D). Hypotonia was initially noted at birth. Acquisition of psychomotor milestones was delayed: the patient reached head control at 4 months of age, sitting position at 7 months, and he started walking with support at 2 years of age and independently at 2 years and 6 months. He started vocalizing from 5 months of age, language appeared at the age of 2 years (bilingual exposure) and was characterized by single words. He has a quiet disposition, with a good degree of interaction with others and with the surrounding environment. Sleep disorder has never been reported.

Brain ultrasound at birth described the presence of regular midline structures, but ectasia of the lateral ventricles at the level of the trigonal area and temporal horns was observed. Choroid plexus of the lateral ventricles appeared irregular and a hyperechogenicity at the temporal periventricular area was noted, more pronounced on the left side; Sylvian fissure in the coronal sections appeared to be wider than normal. Moreover, a rudimentary organization of the insula bilaterally and an inadequate gyration pattern for the age was described. A suspicion of a neuronal migration disorder was made after these findings.

Echocardiogram revealed a patent foramen ovale and a patent ductus arteriosus. Other instrumental tests (abdominal ultrasound, ultrasound of the kidneys and urinary tract, audiometric examination, ophthalmological evaluation) were normal.

### 3.2. Neuroradiological Imaging

Brain MRI was performed for a better characterization of the brain malformation described at birth: it revealed a type 1 lissencephaly with prevalent temporo–occipital–parietal expression and a concomitant double-cortex appearance (Dobyns grade 1–2) (Figure 2). Moreover, the Rolandic areas showed poor differentiation (Figure 2C), with shallow Sylvian sulci and incomplete opercularization and enlarged lateral ventricles in the intermediate-posterior Sector (Figure 2B,C). Partial hippocampal malrotation was also present (Figure 2A). In the SWI (Susceptibility-weighted images) imaging (Figure 2D), prominent and ectopic deep venous circulation was also highlighted, as well as the presence of ectopic periventricular venous structures that has been classified as secondary to the abnormal neuronal migration process.

### 3.3. EEG

Serial EEG showed abnormal background activity with posterior medium-voltage slow waves and rare centro-temporal sharp waves. During NREM sleep, a continuous alpha-like pattern appeared, predominantly in the fronto-centro-temporal area. No clear epileptic seizures have ever been reported and no antiepileptic therapy has been necessary to date.

### 3.4. Genetic Analysis

In order to identify a molecular cause underlying our patient’s clinical features, CGH-array analysis was performed at birth, and was found to be normal. This investigation was followed by molecular analysis of *LIS1* and *DCX* genes (Sanger sequencing and Multiplex Ligation-dependent Probe Amplification), which revealed no alterations. A targeted Next-Generation Sequencing (NGS) custom panel of 182 genes associated with brain malformations (Appendix A) provided no significant result. In order to account for the possibility of low-level somatic mosaicism in genomic DNA derived from blood [28,29], a further NGS panel of 27 lissencephaly/polymicrogyria associated genes, partially overlapping with the previous one and also including *LIS1* [2], was tested and found to be normal. Based on the molecularly unexplained phenotype, the patient was enrolled in the “Undiagnosed Patients Program” of the Ospedale Pediatrico Bambino Gesù (Rome, Italy). WES performed on the trio (proband and parents) revealed a heterozygous variant located at the splice donor site of exon 2 of the *CEP85L* gene: NM_001042475.3:c.232+1del. The variant had not been previously reported in the scientific literature, Human Gene Mutation Database (HGMD) or reference population database gnomAD v2.1.1. It was predicted in silico to alter RNA splicing by abolishing the canonical donor site of exon 2 [23,24,25,26,27], and no legitimate alternative cryptic splice sites were detected. cDNA analysis confirmed the presence in the patient’s cells of an alternate transcript with skipping of exon 2, which was not observed in normal controls (Appendix A). According to the ACMG guidelines [17], the variant was classified as pathogenic.

## 4. Discussion

Neuronal migration is an extremely complex process that requires fine coordination of multiple genes during brain development. These genes mainly encode microtubule or microtubule-associated proteins, responsible for the formation of the cytoskeleton structures that drive the correct cellular migration during nervous system development. Recently, the *CEP85L* gene has been identified as involved in microtubule organization [30], and it has been indicated as a cause of predominantly posterior-expressed lissencephaly [9,10]. In a cellular model, Kodani et al. in 2020 showed that, during the G phase of the cell cycle, the gene product of *CEP85L* is localized in a region surrounding the centriole and that its activity is strongly intertwined with other genes involved in the regulation of neuronal migration, such as *LIS1* [9]. They demonstrated that CEP85L promotes the localization and activation of CDK5 in the centrosome to form a dynamic microtubule cytoskeleton, necessary for neuronal migration in the developing cortex. This latter observation could explain the prevailing posterior distribution of the pathological findings observed in patients.

Furthermore, in the 7 patients reported in their work, Kodani et al. also highlighted the simultaneous involvement of other brain structures, in particular, the recurrence of dysmorphic corpus callosum. In all cases, alterations involved the occipital, parietal and temporal regions, mostly sparing the regions placed rostrally to the central sulcus [9] (Appendix A).

At the same time, variants in *CEP85L* were identified through WES by Tsai et al. in a group of 13 patients with posterior lissencephaly, in the absence of other known molecular causes, *LIS1* included [10]. All cases reported had the characteristic posterior distribution of lissencephaly, which ranged from complete agyria to SBH. The authors defined the level of neurodevelopmental delay and the presence of seizures in each patient and noted that there is no close correlation between the severity of lissencephaly and the intellectual outcome of the individual. Many patients are described as having a normal IQ, indicating a possible transmission to children from an affected parent; this is in contrast with patients with *LIS1* variants, who usually have reduced or absent reproductive fitness as a consequence of the severe clinical condition.

It is also interesting to note that all identified *CEP85L* missense variants fall within a stretch of about 15 amino acids encoded by exon 2, leading to the hypothesis that this may be an essential region for the interaction with other proteins. These missense variants are associated with a variable phenotype, mostly mild (SBH), with the exception of p. (Asp65Asn), which was identified in a patient with a more severe neuroradiological and clinical picture.

The variants associated with the most severe MRI findings are those involving the first codon of *CEP85L*, which determine a marked decrease in protein expression, probably secondary to a start-loss effect. Notably, a second cluster of mutations fall downstream of nucleotide 232, corresponding to the donor splice site of exon 2. We applied to this cluster multiple in silico tools (NNSPLICE, NetGene2, MaxEntScan, SPiCE) and observed that a significant decrease in the efficiency of the canonical splice site was predicted for all variants (Appendix A). Maximum Entropy Models [23] also indicated that the two variants at position +1 are expected to have the strongest effect, while the variant at position +3 is the most likely to allow a fair degree of residual wild-type transcript. This might explain why the six patients with variants at positions +3 to +5 have a generally mild clinical phenotype, while the two cases with variants at position 232+1 (patient VII reported by Tsai et al. [10], with a G>T transversion, and our patient, with a single nucleotide deletion) show a more marked and complex picture of agyria. RNA sequencing performed by Tsai et al. on the two individuals with variant c.232+5G>A indicated leaky skipping of exon 2, leading to an in-frame deletion in about 10% of the transcripts, and did not report other forms of alternative splicing [10]. The other variants are expected to behave in a similar manner. Our in silico analysis reported no cryptic 5′ splice sites nearby, and by cDNA amplification and direct sequencing, we confirmed that the c.232+1del variant also leads to in-frame skipping of exon 2 (Appendix A). Although it has not been verified at the peptide level, this would generate a p.Gly25_Glu77del protein missing the critical region usually affected by the missense variants. The hypothesis of a dominant-negative pathogenic mechanism has been formulated for this in-frame variant [10], but further studies on the protein product would be necessary to elucidate this aspect.

In our patient, the regions of altered gyrification also present with a double-cortex appearance, a combination that has never been reported before in *CEP85L*-related lissencephaly. However, the degree of cortical malformations is not associated with a correspondingly severe EEG and clinical outcome, but only with a moderate neurodevelopmental delay, and in our patient, with the absence of manifest seizures. Unlike other conditions in which lissencephaly is present, such as *LIS1*-related cases, the neurological and intellectual outcome associated with *CEP85L* variants is widely variable, and patients may even show a normal or a slightly delayed development. The main factor influencing the evolution of this condition is represented by the possible onset of epilepsy, which in some cases is scarcely controllable through specific drug therapy and can even lead to a regression of the abilities previously acquired by the individual.

Our patient, despite marked agyria, presented with satisfactory intellectual and motor development, albeit slightly delayed compared to normal. Finally, neither clinically manifest nor electric seizures have ever been observed. The continuous alpha-like EEG pattern typically related to lissencephaly appears only during NREM sleep, and is predominant on the centro-temporal area, which shows less marked anatomical alterations compared to the posterior regions.

This report, by comparing the phenotypic and molecular findings with those of other known cases, allows a further definition of the clinical and radiological characteristics of posterior lissencephaly caused by variants in *CEP85L*. In particular, it underlines the importance of carefully evaluating the EEG pattern, which can present with a typical trait (alpha-like) associated to an atypical distribution and that may not correlate directly to the severity of the anatomical anomaly. Furthermore, it is useful to look for the possible presence of other anatomical anomalies (such any dysmorphism of the corpus callosum or partial hippocampal malrotation) and take into account that the subsequent neurodevelopment can have a favorable evolution despite the presence of relevant MRI findings. This is an aspect to consider, since it can be of comfort to parents of affected children. Our findings also corroborate the presence of a hot-spot region of pathogenic variants located at the donor splice site of exon 2.

Further data could be useful in order to better define *CEP85L*-related lissencephaly, and could derive from the re-analysis of the WES data of patients with radiological features of posterior lissencephaly and no molecular anomaly identified in other genetic loci.

## Figures and Tables

**Figure 1 genes-12-01208-f001:**
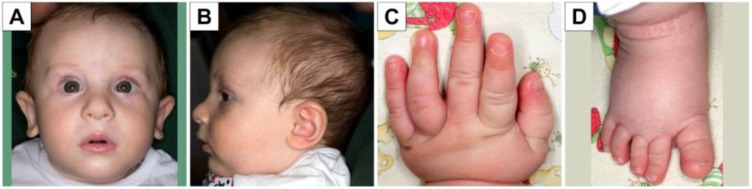
Phenotypic characteristics of the patient. Note bitemporal narrowing (**A**), slightly protruding ears (**A**,**B**) with up-lifted lobes (**A**), posterior plagiocephaly (**B**). Normal hands and nails (**C**,**D**) except for large first toe (**D**).

**Figure 2 genes-12-01208-f002:**
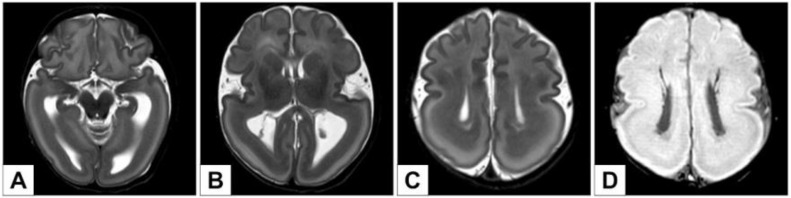
(**A**–**C**): Axial T2-weighted images. Type 1 lissencephaly (temporo–occipital–parietal prevalence) with subcortical band heterotopia (“double cortex”) (**A**–**C**). Rolandic regions poor differentiation with shallow Sylvian sulci and incomplete opercolarization (**C**). Bilateral partial hippocampal malrotation (**A**). Enlarged lateral ventricles in the intermediate-posterior sector (**A**,**B**). (**D**) SWI images (MinIP) of prominent and ectopic deep venous structures in periventricular regions.

## Data Availability

The data that support the findings of this study are available from the corresponding author upon reasonable request.

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
