# Peer review of "Posterior Lissencephaly Associated with Subcortical Band Heterotopia Due to a Variation in the CEP85L Gene: A Case Report and Refining of the Phenotypic Spectrum"

_genes, 2021, doi:10.3390/genes12081208_

Round 1

Reviewer 1 Report

This manuscript reports a novel variant of the CEP85L gene in a patient with posterior predominant lissencephaly, accompanied by subcortical band heterotopia, mild dysmorphic features, and slightly delayed psychomotor development.

Only a small number of patients with this gene mutation have been reported, and the genotype-phenotype correlation is not established. Thus this article adds valuable knowledge to the clinical and research field.

I have only a few comments otherwise it is interesting.

Comments:

  1. What effect does the specific variant “NM_206921:c.232+1del” have on the gene expression? The discussion on the genetic effect of the current variant is not enough.

  1. The reference [7] in line 62 should move to line 59.

  1. Line 156. “opercolarization” should be corrected to “opercularization”.

  1. Line 175. What does anticomitial therapy mean?

  1. Lines 196, 197. “microtubulin” should be corrected to “microtubule”.

Reviewer 2 Report

The paper by Contro et al. submitted for publication at Genes is, generally speaking, well written, concise and straightforward. Regarding its scientific content I provide comments for the genetics part of the study, as I have no expertise to judge the clinical (neurological/imaging) aspects of the patient's examination, which I am sure was undertaken to the greatest extent.

The manuscript, although intended for Genes, provides limited genetic info. I think certain sentences and procedures require further explanations having in mind those readers interested in the genetics arm of the study. For instance, how did the authors define/design the NGS panel of 182 genes for cortical malformations, is it an in-house design? based on data selected from where/which database?

On the same aspect, the authors state “A further NGS panel of 27 genes were analyzed to exclude somatic mosaicisms and were found to be normal.” The same question, as above, applies here. Please provide clarification as to the meaning of “somatic mosaicisms” when analyzing blood-derived genomic DNA, why was this deemed necessary?

What is of importance for me when identifying and reporting a novel, likely pathogenic, mutation is to go a bit beyond the obvious. I am not suggesting extensive functional experiments for this paper, but since there was peripheral blood available, a basic mRNA-cDNA analysis would have been welcome (if CEP85L is expressed in blood). A semiquantitative RT-PCR on agarose gel could show if the mutant c.232+1delG mRNA (expected to be ~160bp shorter than the wild type) is detected in WBCs or not. If mutant cDNA becomes detected, then its (Sanger) sequencing could help explore whether the mutant mRNA is preserved and not degraded. Moreover, the existence and use of a cryptic splice donor site (leading perhaps to intron retention) should be, at least, discussed. Then, the authors could have (some) support to claim a potential dominant-negative effect instead of haploinsufficiency. Experimental data on protein level only (mutant vs wild-type) would prove this claim.

Please bear in mind that in their Neuron paper, Tsai et al. state (I emphasize in bold): “We identified nearby splice-site variants c.232+1G>T and c.232+3G>T in individual VII and an affected father-daughter pair (VIII-a and VIII-b), respectively. Both variants are also predicted to cause a loss of the splice donor site from exon 2 (Desmet et al., 2009). Although we were unable to evaluate RNA, we hypothesize that these variants will result in skipping of exon 2, similar to c.232+5G>A.” .... and they continue ….”The predicted protein products of the c.232+5G>A, c.232+3G>T, and c.232+1G>T variants [p.(Gly25_Glu77del)] should also disrupt this region of the protein while leaving downstream domains intact.” However, Figure S3 in Tsai et al. (with a legend text that is very problematic), provides experimental data only for c.232+5G>A and shows that exon 2 is skipped in ~10% of transcripts in patient IX-a. So, reading it the other way around, it means that 90% of the main transcript likely remains intact, unless Tsai et al. would have shown that this 10% of mutant transcript is sufficient to cause a dominant-negative effect on protein level, which I believe was not shown. Again, experimental data on protein level only (mutant vs wild-type) would prove this claim.

In the second section “Materials and methods”, I would have liked to see a reference for the in silico tools you used for the prediction of the c.232+1delG effect on splicing. You do mention those at the end of the “Results” section, but it is not clear to this reviewer if your team performed the predictions or if you are referring to published data. Also, I saw no discussion regarding the potential of a cryptic splice site becoming activated due to c.232+1delG (hence the possibility of intron retention, or alternative 5’ ss activation) and it is not clear to me if you performed in silico prediction for the existence of cryptic splice sites.

Also, on the same issue, in the “Results” section you cite references 23, 24 and 25, but in the “Discussion” section (lines 238-242) you cite references 24 and 25 (Ref. 23 is missing) and in addition references 26 and 27 – is this a general discussion about the effect of +1 and +3 splicing-affecting variants or is it specific to CEP85L? Please rephrase accordingly to clarify the point to readers.

Minor comments:

Pg 2, line 84, please correct “manufaturer’s”

Pg 2, line 88, please correct “analyzes”

Pg 2, line 90, please correct “manufacture’s”
